# Targeted Cytokine Delivery for Cancer Treatment: Engineering and Biological Effects

**DOI:** 10.3390/pharmaceutics15020336

**Published:** 2023-01-19

**Authors:** Vladislav S. Rybchenko, Teimur K. Aliev, Anna A. Panina, Mikhail P. Kirpichnikov, Dmitry A. Dolgikh

**Affiliations:** 1Bioengineering Department, Shemyakin & Ovchinnikov Institute of Bioorganic Chemistry, Russian Academy of Sciences, 117997 Moscow, Russia; 2Department of Chemistry, M.V. Lomonosov Moscow State University, 119234 Moscow, Russia; 3Department of Biology, M.V. Lomonosov Moscow State University, 119234 Moscow, Russia

**Keywords:** immunocytokines, targeted delivery, fusion proteins, cytokines, antibodies, cancer treatment

## Abstract

Anti-tumor properties of several cytokines have already been investigated in multiple experiments and clinical trials. However, those studies evidenced substantial toxicities, even at low cytokine doses, and the lack of tumor specificity. These factors significantly limit clinical applications. Due to their high specificity and affinity, tumor-specific monoclonal antibodies or their antigen-binding fragments are capable of delivering fused cytokines to tumors and, therefore, of decreasing the number and severity of side effects, as well as of enhancing the therapeutic index. The present review surveys the actual antibody–cytokine fusion protein (immunocytokine) formats, their targets, mechanisms of action, and anti-tumor and other biological effects. Special attention is paid to the formats designed to prevent the off-target cytokine–receptor interactions, potentially inducing side effects. Here, we describe preclinical and clinical data and the efficacy of the antibody-mediated cytokine delivery approach, either as a single therapy or in combination with other agents.

## 1. Introduction

The possibility to boost or inhibit the immune system for treating various diseases has engaged the interest of scientific society for decades. The discovery of cytokines made it real, not only in theory, but also in practice, providing the opportunity to use cytokines for cancer treatment [1,2,3,4,5]. Cytokines are small protein molecules with a molecular weight (Mw) in the range of 5–70 kDa that control immune system responses and normal immune system functioning [6]. Their dysregulation could result in severe autoimmune or chronic inflammatory diseases, including rheumatoid arthritis [7] and asthma [8], or cause pathological reactions, such as a cytokine storm during infections [9]. For this reason, cytokine inhibitors are widely used in clinical practice. Notwithstanding the above-mentioned negative effects, some cytokines play important roles in anti-tumor defense [10] and cell-autonomous immunity [11], or even demonstrate immunosuppressive properties yielding beneficial effects in preclinical animal models. It makes them particularly attractive for the medical treatment of various pathological conditions and diseases. However, applying cytokines in therapy is limited in light of their toxicity [12,13,14,15,16,17], life-threatening side effects, or cytokine release syndrome [18,19]. Overcoming these difficulties would improve their therapeutic efficacy and open up new applications for cytokines.

A promising approach to cytokine toxicity reduction is their site-specific delivery, which could be accomplished either by cytokine gene therapy or antibody–cytokine fusion proteins. The major application field of cytokine gene therapy is cancer treatment, but it has some serious limitations. Direct injections of DNA vectors carrying cytokine genes are restricted by only easy-to-access tissues, such as skin and muscles. Other tissues may be reached using viral vectors, but this method is limited by the efficiency of solid tumor transfection, the immunogenicity of the vectors, safety terms and cost factors [20]. Due to its complexity, gene therapy is usually anticipated by standard radio- and chemotherapy protocols. This type of treatment exerts suppressive effects on the immune system and decreases the effectiveness of cytokine application [21]. Quite a different strategy with a simpler mode of action is represented by the antibody–cytokine fusion proteins, also called immunocytokines [22]. These proteins could be administered subcutaneously or intravenously [23], and they are more accessible for different tissues, show less immunogenicity than genetic vectors, and lack the disadvantages of cytokine gene therapy.

The origin and development of antibody engineering technology allowed for constructing immunocytokines, which consist of two parts; one of them is represented by a cytokine molecule linked to a monoclonal antibody or its fragment by a peptide linker. Due to the specificity derived from antibodies, immunocytokines are targeted preferentially to cancer or immune system cells. This design makes the delivery of cytokines safer as compared to cytokine use in general and diminishes the number and severity of adverse effects [24]. Adding Fc-moiety increases the plasma half-life in comparison with a non-fused cytokine, prolongs its effect, and improves its efficacy [25]. The therapeutic potency of immunocytokines depends on many factors: the molecular format, fused cytokine, antibody target, size of the molecule, and combinatorial properties of other pharmacological agents used in co-therapy. The potential benefits to be brought to clinical practice obviate the necessity to develop immunocytokines, making it a pressing issue for biopharmaceutical research.

## 2. Molecular Formats: Structures and Properties

Monoclonal antibodies are the most often used proteins to construct immunocytokines since they demonstrate high affinity, selectivity, and well-predicted and studied properties and characteristics. They consist of two heavy (Hc) and two light (Lc) chains forming two Fab arms (V_L_C_L_V_H_C_H1_) and a Fc-region ([C_H2_C_H3_]_2_). The main function of the Fc-region is to interact with proteins of the complement system and Fc receptors, including neonatal Fc receptor (FcRn), which plays an important role in antibody recirculation and maintaining a long serum half-life [26,27]. A wide variety of existing antibodies with a broad range of actual and possible therapeutic targets made antibodies extremely convenient objects for devising immunocytokines. Most antibody–cytokine fusion proteins can be classified into three groups: full-scale IgG fusion proteins and intermediate- and small-size molecules based on antibody fragments with binding activity (Figure 1). One of the most common format choices, in case one wishes to achieve the long serum half-life, is a recombinant full-length IgG antibody with a C-terminus-linked cytokine. This recombinant molecule retains all IgG inherent activities, but is bivalent against cytokine receptors, which could be adverse if cytokine–receptor interactions are too strong. This may lead to immunocytokine trapping by the excess of receptors and the resulting lack of necessary concentrations at the target site [28]. One way to overcome this problem is to introduce knob-into-holes [29] or other heavy chain heterodimerization platform mutations into heavy chain genes in order to retain the bivalency and avidity of the antibody, but to decrease the number of cytokine receptor binding moieties to the number of one. Another solution is to switch to smaller formats, such as Fab (antigen-binding fragment) or scFv fragments (single-chain variable fragment). However, antibody fragments are characterized by different pharmacokinetic properties with shorter serum half-life and increased diffusion capacity if compared to the full-scale IgG [30,31]. On the contrary, rapid clearance and diminished tumor retention are the drawbacks of intermediate- and small-sized formats, which cause low absolute accumulation about one fold lower than full-length IgG. By contrast, the decrease in serum half-life may be beneficial in terms of keeping blood concentrations low to reduce side effects.

Notably, some cytokines tend to form dimers (IFN-γ, IFN-β, IL-12) or trimers (TNF-α) [32,33,34], which makes applying the full-scale IgG inconvenient for antibody–cytokine fusion protein construction since the molecular weight of those final products is extremely high. To avoid it, these antibody fragments may be used instead of full-scale IgG, which often results in intermediate-size immunocytokines. Moreover, dimeric and trimeric cytokines may be used to increase the number of antigen binding sites of the intermediate-format immunocytokines and to improve their tumor-targeting characteristics. If the cytokine of choice has a monomeric structure, e.g., IL-2-scFv [35], the number of paratopes may be altered by choosing bivalent antibody-fragment-based formats, such as diabodies [36] or (scFv)_2_ (Figure 1).

Although most of the developed immunocytokines could be classified into the above-mentioned formats, some researchers attempt to experiment with immunocytokine structures in order to improve their characteristics and properties, in particular, to decrease their toxicity and to increase their capacity to penetrate tissues. The smallest immunocytokines consist of only VHH (single-domain antibodies or nanobodies) or one scFv fragment fused to the cytokine. The small molecular weights of VHH [37,38] and scFv-based immunocytokines make them suitable for overcoming high interstitial pressure and increases the tumor tissue penetration, as it is known that the tissue-penetration capability of particles highly depends on their size [39,40]. Using the small-size formats provides an opportunity to reach the core of small tumors in the case of other contraindicated treatments.

Some immunocytokines are designed to reduce the toxic effects of the cytokines by blocking their direct interactions with receptors during transportation through the bloodstream. In general, this approach involves using cytokine neutralizing antibodies, which may be incorporated into mutein proteins through tumor-specific protease-cleavable linkers, or form non-covalent complexes with cytokines. The first type of mutein may be based either on a single polypeptide construct comprising cytokine and a cytokine inhibiting domain connected via a cleavable linker, or on heterodimeric Fc-containing constructs (Figure 1). Despite the structural differences, the linker susceptible to tumor-specific proteases plays the key role in cytokine tumor-specific release in «conditionally activated» or «receptor masked» recombinant immunocytokines [41,42,43]. However, these proteins are devoid of any vehicle for specific delivery, a disadvantage that may be eliminated by using bispecific antibodies. This concept was applied in the construction of the molecular complex of IFN-β and anti-ErbB2 IFN-β-neutralizing bispecific antibody (Tz-B16) for ErbB2+ solid tumor treatment (Figure 1). One arm of the bispecific antibody is responsible for the ErbB2 targeting of the complex, while the second arm forms the antibody–interferon neutralizing complex, which dissociates at the tumor site [44]. Summing up, each of the formats has its own advantages and limitations, making the choice of the optimal structure of the immunocytokine molecule ambiguous and controversial.

## 3. Mechanisms and Modes of Action

Since immunocytokines consist of the antibody and cytokine moieties, their mechanisms of action include the inherent ones typical of both antibodies and cytokines, with some new properties emerging. Full-scale IgG-based immunocytokines retain CDC (complement-mediated cytotoxicity) and ADCC (antibody-dependent cellular cytotoxicity) activities without any significant decrease compared to the prototype antibodies [45,46]. Thus, they directly bind to tumor cells and trigger those reactions. However, the concentrations of immunocytokines used during therapies are one or two orders of magnitude lower than those of anti-cancer monoclonal antibodies, which makes ADCC and CDC as a mechanism controversial and rather additional than principal. Moreover, some immunocytokines undergo site-specific mutagenesis with a purpose to disrupt Fc and C1q binding, and thus improve localization and decrease off-target effects [47]. Being a delivered molecule, a cytokine plays a more important role in antibody–cytokine fusions. Distinct cytokines may activate different subsets of leukocytes and lymphocytes and induce their proliferation or infiltration at the site of the tumor. Pro-inflammatory payloads increase NK- and T-lymphocyte killing capability, but some of them (e.g., IL-2) enhance immunosuppressive properties of T_regs_ (regulatory T cells), mostly an unwanted effect during cancer treatment [48]. Targeted cytokine-mediated activation of the immune cells may trigger massive cytokine production at the disease site, making the local environmental conditions more favorable for local immune system cell functioning.

The delivered cytokines can also directly affect tumor cells and decrease their proliferation and motility or increase MHC I expression and antigen presentation, making a tumor more recognizable for immune cells and less capable of metastatic spread. Some of the cargoes (e.g., TNF-α or TRAIL) can directly induce apoptosis through ligand–receptor interactions, and therefore, could be considered as “magic bullets”. Tumor vasculature is also an important target for cytokines; for example, TNF-α, IL-1, and IL-6 were reported to promote intravascular blood coagulation of small capillaries [49]. Finally, immunocytokines can act similarly to bispecific antibodies, establishing the crosslinks between tumor and immune system cells (Figure 2). Interactions mediated by IL-2-based antibody–cytokine fusion proteins induce polarization, increase cell adhesion, and activate immune synapse formation between tumor and IL-2R+ NK cells, facilitating their cytotoxic functions [50]. Immunocytokines were reported to redirect FcR-lacking cells, e.g., effector T cells, to malignancies through the formation of stable conjugates between cell membranes. After establishing these contacts, T cells mediate Fas-Fas-dependent lysis, followed by consequent tumor cell death [51].

## 4. IL-2-Based Immunocytokines

Interleukin-2 (IL-2) is a 15–16 kDa proinflammatory cytokine known and discovered as a T-cell growth factor [52]. IL-2 has multiple functions: it promotes the growth and development of peripheral immune cells, and it is essential for Th_9_ generation and T_reg_-cell differentiation, but it also demonstrates an inhibitory effect on Th_17_ and T_FH_ differentiation. There are three different classes of IL-2 receptors with different affinities, consisting of the IL-2Rα (K_D_~10^−8^ M) subunit only; the IL-2Rβ and IL-2Rγ subunits (K_D_~10^−9^ M); and the IL-2Rα, IL-2Rβ, and IL-2Rγ subunits (K_D_~10^−11^ M). Whereas intermediate-affinity receptors are expressed mostly on resting NK and T cells, high-affinity receptors are typical of lymphocytes after their activation, making them more sensitive to this cytokine. IL-2 is also able to improve the tumor-killing capacity of different immune system cells by increasing the cytolytic activity of NK cells and lymphokine-activated killer cells and inducing the rapid proliferation of CD8+ T cells [53]. Using IL-2 in clinical practice (Aldesleukin) started three decades ago and was considered as the first effective immunotherapy for human cancer [4].

Further studies demonstrated that IL-2 therapy may be successfully combined with adoptive cellular therapy or genetically activated T-cells administration, resulting in a statistically significant increase of survival rate, as well as other clinical indicators [54]. However, the limited success of IL-2 therapy might be explained by a narrow therapeutic index, vascular leakage syndrome, and toxicities, which make IL-2 more useful combined with chemotherapy [55]. Other facts presented below demonstrate that IL-2 has a few protumorigenic properties, which makes its use for cancer treatment less encouraging. For example, IL-2 is an inducer of Th_9_-cell differentiation and Th_9_-mediated IL-9 production, two factors correlating with better tumor cell survival and apoptosis resistance [56,57,58,59,60,61,62,63,64,65,66,67]. IL-2 inhibits the differentiation of Th_17_ cells, a population of lymphocytes performing an important function in protecting the organism from exogenous pathogenic fungi and bacteria that makes patients less resistant to infections during therapy. IL-2 was also shown to be crucial for the suppressive activity of peripheral and thymus-derived T_reg_ cells, normally responsible for self-tolerance during tumorigenesis stimulation, and infiltration of peripheral T_reg_ into tumors may inhibit antitumor immunity [58]. Finally, IL-2 is characterized by a serum half-life limited to ≈7 min [59]. All the facts we have discussed thus far, as well as other minor factors, led scientists to develop new methods of overcoming IL-2 toxicity, side effects, and limitations.

The restrictions related to limited serum half-life and T_reg_ activation could be overcome by IL-2 engineering, making fusion proteins, or chemical conjugation with PEG. Nonetheless, none of these techniques dramatically increases the specificity of the cytokine to tumor cells [60]. One of a few methods used to overcome the negative immunosuppressive effects of IL-2 is IL-2 receptor–cytokine binding interface engineering performed to decrease the binding to high-affinity T_reg_ interleukin receptors. IL-2 with abrogated IL-2Rα (CD25) binding or enhanced IL-2Rβ binding was demonstrated to induce less efficient T_reg_ expansion and tumor infiltration, with more efficient tumor-specific CD8+ T-cell proliferation [47,61]. A no-α-designed mutant form of IL-2 with reduced binding capacity to IL-2Rα showed a greater therapeutic efficacy and anti-metastatic effect in a mouse metastasis model than wild-type IL-2. This could be explained by the preferential stimulation of NK and CD8+ T cells, along with the avoidance of stimulation of CD4+F FoxP3 T_regs_ [62]. However, both the toxicity and lack of tumor specificity represent a serious limitation for harnessing the engineered IL-2 forms. Therefore, IL-2 and tumor-specific antibody fusions are widely constructed and tested.

Currently, there is a set of IL-2–based immunocytokines already developed or under development. They are to target disialoganglioside GD-2, CEA (Carcinoembryonic antigen), CD-20, FAP (fibroblast activation protein), CA 125, and other cancer antigens [24,63,64,65,66] (Table 1). Besides the antigen specificity, the antibody format and IL-2 form are the main differences and peculiarities, which have a significant impact on the biodistribution and pharmacokinetics of the recombinant molecules. Among the CEA-targeted IL-2-based immunocytokines, cergutuzumab amunaleukin is one of the most developed, with phase 1 clinical trials already completed. It consists of anti-CEA bivalent monoclonal antibody with abolished interaction with FcγR and C1q, fused to the N-terminus of a single molecule of IL-2v variant with restricted IL-2Rα binding [67]. Cergutuzumab amunaleukin (CEA-IL-2v) was designed to increase the therapeutic index, improve the pharmacokinetics of IL-2, accumulate at CEA-expressing tumors, and induce local immune response, as well as reduce binding to CD25-expressing endothelial cells and avoid predominant activation of T_regs_ and undesired Fc effector functions [47,68]. The superior anti-CEA antibody avidity (K_D_ = 390 pM), compared to the affinity of IL-2 moiety to intermediate-affinity receptors (K_D_ = 40 nM), prevents the sequestration of circulating immunocytokine because of the retention by IL-2R-expressing cells. This improves the distribution of CEA-IL-2v and its tumor uptake. In contrast to the CEA-IL-2 wild-type form, CEA-IL-2v in animal models induces preferential expansion of CD8+ T and NK cells in blood, lymphoid tissues, and tumors and increases the number of tumor-infiltrating immune cells. The efficacy of the monotherapy by CEA-IL-2v was tested in C57BL/6 mice injected with MC38-CEA or PancO2-CEA (CEA-positive) cells. Animals treated with 2 mg/kg and 0.5 mg/kg of CEA-IL-2v for MC38-CEA or PancO2-CEA tumors, respectively, showed a significant increase in median survival of 63 vs. 42 days and 42 vs. 30 days, respectively, without any dramatic toxic effects registered. Even more impressive results were obtained during a combination of PD-L1 checkpoint blockade with CEA-IL-2v in PancO2-CEA intrapancreatically injected mice. An increase in the median survival for vehicle-treated, CEA-IL-2v-treated, and concomitant dosing of CEA-IL-2v with PD-L1 blockader was observed from 29 to 43 and 57 days, respectively. Considering that CEA-IL-2v is ADCC-deprived, its use in combination with ADCC competent antibodies eliminates this shortcoming by regulating the ratio of cytokine and ADCC-carrying moieties. This assumption was confirmed in human CD16-transgenic SCID mice. A pair combination of CEA-IL-2v and three human IgG1-isotype antibodies—trastuzumab, cetuximab, and imgatuzumab—demonstrates superior efficacy and anti-tumor effects compared to the monotherapies [47,69,70]. Clinical data show CEA-IL-2v tumor-specific accumulation with a non-significant dose-dependent trend accompanied by hepatic and splenic uptake from the first screening measurements. Notably, CEA47-IL-2v accumulates both at CEA+ and CEA- tumors, but with a more pronounced effect for CEA+ tumors. How exactly CEA-IL-2v accumulates at CEA- tumors remains unclear, but IL-2R may play a crucial role in this process [71].

Despite the considerable progress in cergutuzumab amunaleukin development, its IL-2-based applications are limited only by one tumor marker. This gap is being filled with other perspective immunocytokines discussed below. Two of them, L19-IL2 (Darleukin) [73,76], targeted to fibronectin extra-domain B, and F16-IL2 (Teleukin) [77], aimed at Tenascin C, have already completed Phase I clinical trials and are currently in Phase II. L19-IL-2 and F16-IL-2 are diabody format immunocytokines composed of two ScFv fragments with C-terminus-fused IL-2, which means the delivery of two cytokines by each molecule [72,77]. During preclinical studies, L19-IL-2 demonstrated an excellent tumor–to–blood ratio of 33:1 24 h post injection, and the ratio between the percentage of injected dose per gram of tumor and other organs exceeded the value of 10. L19-IL-2 also has a vastly improved therapeutic index compared to cytokine or IL-2-fused diabody composed of ScFv fragments of irrelevant specificity [64]. The aforementioned information and improved therapeutic index make using L19-IL-2 and F16-IL-2 safer and possible for cotreatment with other therapeutics. L19-IL-2, combined with rituximab and CTLA-4 blockade or L19-TNF immunocytokine, was able to completely eradicate β-cell lymphoma xenografts and demonstrated a substantial anti-tumor effect, even when used as a single agent. Depletion experiments revealed that the key role in antitumor activity was mediated by NK cells [78]. For L19-IL-2 protein-based combination therapies, the vaccination effect was further observed: cured mice did not develop any tumors after being re-challenged by implantation of the same lymphoma cells [64]. Immunocytokines may also be used with practiced chemotherapeutic agents that improve their anticancer potency. F16-IL-2, an anti-Tenascin C immunocytokine, while being co-injected with doxorubicin (4 mg/kg) or paclitaxel (1 and 5 mg/kg), dramatically increased the median survival time in mice bearing breast cancer xenografts and showed a synergistic effect [79]. Moreover, glioblastoma xenografts-challenged animals after treatment with F16-IL-2 and temozolomide stayed tumor free for 160 days before being sacrificed. Biodistribution studies witnessed tumor-specific accumulation of 4% ID/g 24 h post injection, and an increase in leukocytes and NK-cell tumor infiltration [80].

Quite a different tumor marker was selected during Selectikine (NHS-IL-2_Lt_, EMD521873) development, which consists of an IL-2 low-toxicity mutant (D20T) fused at the N terminus to a humanized NHS antibody targeted against DNA and histone complexes released at the necrotic core of tumors [81]. In syngeneic mouse metastatic model experiments, intravenous administration of 80 mcg/injection of NHS-IL-2_Lt_ for 5 days [63] dramatically reduced the metastatic load in lungs and liver. NHS-IL-2_Lt_ has already completed Phase I of clinical studies and demonstrated preliminary data of inefficacy [24]. Patients with different types of cancer tumors (colorectal, ovarian, prostate, and renal cancers) were divided in groups, and Selectikine at 0.15, 0.3, and 0.45 mg/kg was administered intravenously every 3 weeks during 3 consecutive days. The treatment was followed by a strong activation of T cells and a minor activation of NK cells. Unfortunately, only disease stabilization was observed in a few patients. However, these results may have been obtained due to the heterogeneity of the tumors in the patient groups and the considerable influence of prior treatment [74,82].

## 5. IL-12-Based Immunocytokines

IL-12 is a 70 kDa pro-inflammatory cytokine modulating NK- and T-cell activities. Its biologically active form is produced by monocytes, neutrophils, dendritic cells, and B cells in response to pathogen stimulation. This form comprises p35 and p40 proteins linked by a disulfide bridge. Anti-tumor and antimetastatic activities of IL-12 were exploited in several preclinical models [83], but during clinical studies, IL-12 administration failed to elicit any sustained anti-tumor response because of several toxicities, adverse effects, and the increase in tumor immunosuppressive properties by IL-10 induction [84]. In light of the above, the untargeted administration of IL-12 is not pursued nowadays or in the future, but targeted delivery of IL-12 combined with other therapeutics seems to be rather promising. Experiments performed on mouse models demonstrated that tumor-specific delivery of IL-12 increases the number of T lymphocytes, macrophages, and NK cells infiltrating the tumor and decreases the angiogenesis [85]. According to Pasche and coauthors, the format of immunocytokine greatly impacts the effectiveness of the fusion protein if the cytokine molecular weight is high or it consists of two or more subunits, e.g., IL-12 [86]. After failing to localize an F8-based IL-12 diabody-based immunocytokine (average M_w_ 190 kDa) at the tumor site, they decided to use a ScFv platform for fusing the construction. Two recombinant proteins made up of identical scFv fragments and p35 or p40 subunits (Figure 1) were used for IL12-F8 heterodimer formation (scFv_F8_-p35/p40-scFv_F8_) in order to localize IL-12 at fibronectin extra domain A-expressing tumors. In a mouse model with animals bearing F9 large subcutaneous tumors (150–250 mm^3^) among the tested molecules only, IL12-F8 was able to mediate significant growth retardation after a single-dose injection (6 mkg). Immunohistochemistry analysis of the tumors revealed a massive tumor infiltration by NK cells and leukocytes and a mild increase in CD4+ T cells. During experiments, animal serum levels of IFN-γ were altered, which is important for tumor angiogenesis inhibition by CD4+ T cells [87]. Better anti-tumor effects were observed during co-administration of IL12-F8 with chemotherapy. A combination of IL12-F8 with paclitaxel gave encouraging results, with complete tumor eradication in all animals in A20 lymphoma models, and achieving 50% tumor-free animals in the tested sample in F9 teratocarcinoma models [86]. Another immunocytokine, L19-IL12, which is specific to fibronectin EBD, demonstrated an anti-tumor effect very similar to IL12-F8. L19-IL12 is an L19 scFv-based fusion protein containing both p40 and p35 interleukin subunits separated by the flexible linker sequence. In C51 colon-carcinoma-bearing mice, 2.5 mcg injections of L19-IL12 every 48 h for 13 days suppressed tumor growth in the experimental group to the weight of 0.31 g, compared to 1.45 g in the control group. Even superior inhibitory effects were obtained in a F9 murine teratocarcinoma model in 129Sv mice in animals treated with L19-IL12, with the final tumor weight equal to 0.11 g versus 1.45 g in the non-treated group. Further experiments proved the favorable tumor-targeting properties of L19-IL12 recombinant proteins with a tumor–to–blood ratio over 31 and an apparent serum half-life of nearly 22 h [88].

Nowadays, only two IL-12-based immunocytokines have been progressed to clinical trials (Table 1): NHS-IL12 and BC1-IL12. NHS-IL12 is a full-scale IgG with a C-terminus-fused IL-12 p35 subunit via the glycine–serine linker. The other subunit p40 is translated separately and forms a heterodimer with p35 by a disulfide bond. Compared to systemic administration of IL-12, NHS-IL12 demonstrated reduced toxicities due to a targeted delivery of the molecule to the necrotic core of the tumor. A murine version of NHS-IL12 (NHS-murIL12) exerts moderate dose-dependent anti-tumor effects on subcutaneous tumor growth, even after a single injection in animal models. Animals treated with NHS-muIL12 show increased MHC I expression on dendritic cells and proliferation of CD49B+ NK, and CD8+ T cells display stronger p15E-specific CD8+ T-cell response and altered IFN-γ serum levels. Despite the difference in the molecular weights of IL12 and NHS-muIL12, the latter demonstrated a tumor-retention effect superior to recombinant muIL-12 [89], which enhanced and prolonged cytotoxic effector functions of CD8+ T cells at the tumor site. The cured mice did not develop any tumors after a repeated administration of the same tumor cell line, which indicates the vaccination effect of the treatment. A fundamental advantage of NHS-muIL12 is an increase in local tumor IL-12 concentration that seems to be important to bypass the immunosuppressive tumor microenvironment and fully activate tumor-specific T cells [24]. A combination of NHS-IL12 and anti PD-L1 antibody avelumab demonstrated a synergistic anti-tumor effect in MC38 and MB49 tumor-bearing C57BL/6 mice and seems to be promising for tumor cotreatment [90]. Currently, NHS-IL12 has already completed phase I clinical trials, and it was consistently well tolerated, with preclinical data suggesting its potential to improve anti-tumor responses with other standard treatments [91,92]. Another immunocytokine, BC1-IL12, finished phase I clinical trials for renal cell carcinoma and malignant melanoma treatment and proceeded to clinical trials phase II. BC1-IL12 is a recombinant protein consisting of humanized BC1 mAb targeting EDB-containing fibronectin (different epitope from L19) [92] and a C-terminus-fused p35 IL-12 subunit with a representative structure similar to NHS-IL12. During experiments in xenogeneic mice with subcutaneous tumors, BC1-muIL12 succeeded in decreasing the number of metastases. In a PC3 lung cancer model in SCID mice lacking any functional T and B cells, 16 mcg injections of BC1-muIL12 daily were enough to completely prevent further tumor outgrowth [93]. The maximum tolerated dose (MTD) in cynomolgus monkeys determined for 8 weeks was 2.5 mcg/kg, which is 10-fold higher than the corresponding value for huIL-12. In human patients, the MTD was defined as 15 mcg/kg, and the serum half-life was approximately 22 h. During clinical studies, some grade 2 toxicities were registered, but, overall, were lower than those reported for IL-12 as a monotherapy agent [94,95].

## 6. TNF-Based Immunocytokines

Tumor necrosis factor alpha (TNF-α) was discovered back in 1975 as a factor released from host cells, causing hemorrhagic necrosis of sarcoma Meth A and other tumors [96]. TNF-α is primarily synthesized by immune cells as a 34 kDa transmembrane protein with a 17 kDa extracellular domain. After being cleaved by TACE/ADAMS17 proteinase, its soluble form is released and forms homotrimers capable of interacting with TNF receptors [97]. The roles of TNF-α in carcinogenesis are quite controversial. On the one hand, TNF-α promotes inflammation, regulates cell survival and immunosuppression, and can induce invasion of neoplastic cells into the surrounding tissues [98]. All these processes corelate with poor prognosis and ultimately complicate cancer treatment. On the other hand, TNF-α responsible for programmed cell death activation exerted promising anti-tumor effects in several experiments [99]. Considering the above, it is most logical that TNF-α is being investigated as a potent anti-tumor agent. However, high toxicity is a serious limiting factor, which imposes restrictions on the systemic use of this cytokine, with limb perfusion as a single effective method [100].

Taking into account the satisfactory tumor–to–blood ratio of many immunocytokines, the appearance and development of TNF-based antibody fusion proteins seems only natural. Because of the trimeric structure of active TNF-α, using full-length IgG will lead to the formation of immunocytokine complexes with extremely high molecular weights above 500 kDa, with a very limited diffusion capacity. That is why antibody fragments (Fab or scFv) and minibodies are the most convenient vehicles for TNF-α delivery. A fully human L19 antibody is one of the most often used molecules to construct TNF-based antibody–cytokine fusion proteins. In BALB/C mice with implanted WEHI-164 cells L19-mTNF, an immunocytokine consisting of L19 antibody fragment and murine TNF-α demonstrated mild tumor growth inhibition as a single agent, with complete response in one of four animals. The combination of L19-mTNF and trabectedin enhanced this effect to three of four complete responses. Another chemotherapeutic, decarbazine, as a single agent was capable of eradicating tumors in three out of four mice, but while administered with L19-mTNF, the number of complete responses increased to four of four. Very similar synergistic effects were observed for L19-mTNF and anti-PD-1 antibody [101]. Anti-tumor effects mediated by antibody-based TNF-α delivery could also be increased by a proper combination with other cytokines, e.g., IL-2. Menssen and coauthors reported that in their multiple myeloma model experiments with BALB/C J558L mice, a combination of two immunocytokines, L19-TNFα and L19-IL2, yielded the most effective results, as 75% of animals responded to the therapy, and 58% achieved complete tumor eradication. Treatment with either L19-TNFα or L19-IL2 was less efficacious, leading to complete tumor eradication in 42% or 25% mice, respectively [102]. TNF-based immunocytokines also display a very pronounced cytotoxic effect against the endothelial cells, which may be a cause of tumor vessel lesions. This speculation is confirmed at least by the properties of TA99-TNFα, an immunocytokine that consists of a TA99 scFv fragment (anti gp75 melanoma antigen) and TNF-α moieties. TA99-TNFα was able to kill fibroblasts and endothelial cells, but displayed minimal activity against B16 melanoma cells, in contrast to TA99 IgG2a antibody. In vivo TA99-TNFα injections (7 mcg) boosted the influx of NK cells and macrophages into B16 lesions and caused tumor necrosis. According to Murer [103], a combination of TA99-TNFα and TA99 antibody could be an effective remedy, either for tumor treatment or metastasis prevention. Another immunocytokine, F8-TNFα, an analogue of L19-TNFα, was also reported to induce intravascular coagulation of tumor blood vessels. In a sarcoma mouse model, a combination of doxorubicin (5 mg/kg) and F8-TNFα (2 mg) was able to completely eradicate an established 70 mm^3^ tumor, a result unachievable by a single-agent treatment [104]. Despite its high toxicity, L19-TNFα (Table 1) was well tolerated during phase I of preclinical studies, starting from 1.3 up to 13 mcg/kg, causing only mild toxic effects, such as mild chills, nausea, and vomiting. The drug was administered intravenously as three injections per week, and the maximum peak serum concentrations observed were 73.14 mcg/L, and the serum half-life was 33.6 min. With no maximum tolerated dose reached in this study and no objective responses registered, transient stable disease occurred in 19 of 31 patients [105]. Intralesional injections of two immunocytokines, L19-IL2 and L10-TNFα, in the IIIB/C and IVM1a stages of metastatic melanoma patients during phase II clinical studies showed more promising results. In this study, a complete response was observed for 28.3% of lesions and a non-complete response for the rest. Moreover, in patients who demonstrated a complete response, 53.8% of the non-injected lesions also responded to therapy, indicating an immunostimulatory effect [106].

## 7. Interferon-Based Immunocytokines

Initially, interferons were discovered and characterized for their antiviral properties. Meanwhile, in the context of cancer, they exhibit several direct and indirect therapeutic effects [107]. Among all interferons (IFN), only IFN-α, IFN-β, and IFN-γ are exploited for immunocytokine construction. Interferon alpha (IFN-α) is approved by the FDA for treating patients with high-risk melanoma, follicular lymphoma, hairy cell leukemia, and chronic myelogenous leukemia and as the first-line treatment drug for renal cell carcinoma. Using IFN-α in humans polarizes immune responses towards Th1, increases NK cytotoxicity and survival, and enhances the propagation of cytotoxic T lymphocytes and the maturation of dendritic cells. For some cancer types, an anti-angiogenic effect on tumor vasculature and the ability of IFN-α to induce caspase-dependent apoptosis were observed [108]. Unfortunately, systemic administration of IFN-α is limited by a short therapeutic window and the lack of any specificity to tumors and malignancies. In contrast, IFN-α-based immunocytokines demonstrate good targeting properties. For example, fusion of IFN-α to anti-CD38 mAb increased cytokine specificity to CD38+ tumor cells 10,000-fold, meaning that patients might be safely treated with high doses of fusion protein [109]. After injection, IFN-α–based immunocytokines tend to localize at the tumor microenvironment and demonstrate good tumor–to–blood ratios close to 21:1 [110]. Reducing the severity of off-target effects might be achieved by an additional approach aimed at IFN-α engineering. Replacing wild-type IFN-α with an engineered, attenuated version (att) with the capability of inducing proinflammatory markers decreased the toxicity of CD38-(att)IFN-α for animals 100-fold, without substantial loss of anti-tumor activity. Despite using an attenuated version of interferon, anti-CD38-(att)IFN-α sustained significant anti-tumor activity in various cell lines and was capable of completely eradicating the established NCI-H929 subcutaneous tumors in all tested mice [109]. Although IFN-α-based immunocytokines display pronounced anti-tumor and anti-proliferative effects, some data suggest that they might be implemented on tumor vasculature through a direct action. Despite tumor-specific accumulation of scFv(F8)-IFNα in two EDA+ tumor models and a pronounced antitumor effect, immunohistochemical staining revealed the preferential localization of the immunocytokine close to endothelial and perivascular cells at different injection administration doses. In addition, from a safety perspective, scFv(F8)-IFNα was well tolerated in mice, even in repeated 150 mcg injections [110]. In syngeneic mouse tumor models resistant to anti-PD-L1 therapy, IFN-α full-scale IgG-fusion protein infusions, in contrast to IFN-α alone, were tolerated well and were capable of three-fold tumor volume reduction compared to the control group [111]. Notably, these results demonstrate that IFN-α fusion proteins may well be considered for treating patients resistant to anti-PD-L1 therapy. The effectiveness of IFN-α-based immunocytokines strongly depends on in vivo models, one of the most sensitive being the Daudi xenograft model. Immunocytokine 20-2b, comprised of IFN-α2b moieties and CD20-targeting IgGs, shows enhanced ADCC compared to veltuzumab and increases the median survival time in SCID mice in an early Daudi xenograft model by more than 100 days over saline and veltazumab groups given even a single dose [112]. The recombinant molecule (C2-2b-2b), with a similar format targeted at human leukocyte antigen (HLA-DR), demonstrates enhanced apoptosis-prompting ability in multiple myeloma and lymphoma cells, compared to 20-2b and the mixture of anti-HLA-DR mAb and IFN-α2b. In the Daudi model, C2-2b-2b significantly increased the survival time after 1 mcg injections, compared with other tested agents. For both anti-CD20 and HLA-DR immunocytokines, the mice survival correlated with their in vitro cytotoxicity, standing in the subpicomolar IC_50_ range. Although in vivo the benefit of C2-2b-2b was superior to its cognate anti-CD20 fusion protein, C2-2b-2b specificity also caused elevated toxicity against healthy peripheral blood mononuclear cells [113], a fact mentioned as a side effect. A bispecific variant constructed based on these two immunocytokines exhibited an even greater anti-tumor effect in Daudi Lymphoma than its monospecific prototypes [46]. However, the full potential of the conjugates has not yet been demonstrated due to the low sensitivity of murine cells to human interferon.

Interferon beta (IFN-β) is not extensively used for anti-cancer therapeutic development, though some properties of this cytokine imply that it could be used for potent treatment. One of many IFN-β antitumor mechanisms consists of cell S-phase cycle arrest [114]. IFN-β also downregulates c-myc expression, one of few protooncogenes [115]. An increase in Fas, FasL, and TRAIL expression induced by IFN-β may be exploited for targeted cas8-dependent apoptosis mediation [116]. In addition, angiogenesis in tumors is highly suppressed by IFN-β, even at low doses [117]. In vitro studies confirmed that the direct killing effect of IFN-β-based immunocytokines is likely to depend more on interferon activity than on antibody activity [118]. In vivo ErbB2-targeted IFN-β demonstrates considerable tumor growth suppression in humanized mouse gastric cancer xenograft models in a direct manner. Tumor growth inhibition was accompanied by a local 2.5-fold increase in CD8+ T cells and a 2.4-fold decrease in T_regs_ compared to the vehicle control [82]. Interestingly, in vivo IFN-β fusion protein suppresses tumor growth mostly by inducing host immune responses [119]. The half-life of IFN-β within full-scale-based immunocytokines increases from 88 min to 15 h, making it more convenient for administration during treatment. Although in mice [118,119], human IFN-β-based immunocytokines are well tolerated, this conclusion could not be directly extrapolated to humans due to the low sensitivity of the animals to human interferon. Therefore, its toxicity would still be a serious challenge to overcome.

Delivering interferon to a lesion in a safe manner might be achieved in a neutralized state with a few strategies. One of the strategies is to deliver interferon by a bispecific antibody targeted to the tumor cell marker and capable of binding and neutralizing the interferon with a second arm, thus disrupting interferon–receptor interactions during transportation, which may cause potential side effects. Due to high molecular weight, the molecular structure comparable to IgG and the active Fc-part of this complex is stable and might sustain satisfactory serum persistence [44,120]. Another strategy uses latency-associated protein of TGF-β1 fused to IFN-β via a matrix metalloproteinase labile linker, which is responsible for masking interferon from receptor interactions. At the tumor site, matrix metalloproteinases cleave the linker with a following release of interferon in its active form. The ‘latent’ cytokine has a 37.6 times longer half-life than was reported for IFN-β alone and, thus, could be administered systematically at lower doses [43].

Interferon gamma (IFN-γ) is a type II interferon produced by NK cells and activated cytolytic T cells. IFN-γ participates in macrophage activation, increases their oxidative metabolism, and enhances tumor cell killing. Furthermore, IFN-γ is responsible for the augmentation of MHC expression, which enhances the tumor cell immunogenicity [121]. For different types of cancer, IFN-γ-mediated antiproliferative, antiangiogenic, and proapoptotic effects were discovered, and some clinical studies were reported [122,123,124]. IFN-γ targeted to fibronectin EDA, EDB, or CD70 demonstrated superior antitumor activity, compared to non-specific IFN-γ, in distinct models: sarcoma, teratocarcinoma, and lung cancer. The increase in the number of infiltrating macrophages and T and NK cells was registered by immunofluorescence analysis of histological tumor tissue samples in IFN-γ-treated animal groups. Targeted delivery of IFN-γ resulted in tumor microenvironment and metastasis control, compared to non-fused immunocytokine moieties. However, partial immunocytokine sequestration by other tissues containing IFN-γ receptors was observed during experiments. Additional IFN-γ administration resulted in competition for receptors with immunocytokines, and they partially resolved that problem without any apparent influence on animal body weight and negligible toxic effects [125,126,127].

## 8. Other Immunocytokines

In addition to the cytokines listed above, various other ones have been investigated for the construction of antibody fusion proteins. IL-1β and IL-6 fused to anti-fibronectin EDA diabody were constructed and tested in a murine F9 teratocarcinoma model, exhibiting 10% and 5% animal weight loss at 5 and 225 mg dose, respectively. Unfortunately, only 50% tumor growth rate inhibition was observed, which is substantially lower than for other proinflammatory immunocytokines (e.g., F8-TNF-α) [128]. Another cytokine with a potent antitumor effect was used for F8-mIL7-F8 construction, a molecule comprising two anti-fibronectin EDA scFv fragments and murine IL-7. This recombinant protein exhibited good tumor targeting, with tumor–to–blood ratio being 16:1 24 h post injection, and it dramatically inhibited F9 tumor growth in mice. The combination of F8-mIL7-F8 and paclitaxel improved this therapeutic effect, compared to the monotherapy treatment [129]. Tumor-targeting properties of distinct fusion proteins in special cases may greatly depend on the glycosylation profiles of cytokines. A diabody format immunocytokine, F8-IL-9, produced in transient CHO cell culture was capable of localizing at F9 teratocarcinoma lesions, but after establishing the stable cell line, failed to do so. IL-9 has four O-glycosylation sites; thus, the difference in targeting two identical proteins was explained by the difference in glycosylation in stable and transient cell cultures [130]. However, further research of F8-IL-9 showed the futility of F8-IL-9 anti-cancer treatment, but its potency for pulmonary hypertension treatment was discovered [131].

IL-15 plays different roles in immune regulation, but also may act as a proinflammatory cytokine. In cancer models, IL-15 acts in the same manner as IL-2, being less toxic and considered as a promising cytokine for metastasis treatment. In an EL4 GD2+ lymphoma tumor model, anti-GD2-IL-15 significantly inhibited tumor development and increased mice mean survival, whereas administration of anti-GD2 diabody and IL-15 produced no significant effects. The efficacy of recombinant anti-GD2-IL-15 was proved in a liver metastasis NXS2 cell model, where this immunocytokine totally inhibited the development of metastasis. The bioavailability of anti-GD2-IL-15 was higher than that of IL-15, and its serum half-life increased 30-fold [132]. Another IL-15-based immunocytokine, comprised of IL-15 and full-size anti-PD-L1 antibody (srKD033), showed a robust antitumor effect with even a single dose in several diverse syngeneic murine tumor models. In a CT26 murine colon cancer model, a single dose of srKD003 produced durable antitumor immunity and resistance to challenges. The retention of srKD003 in the tumor microenvironment facilitated IL-15-dependent cytotoxic cell expansion, activation of T and NK cells via PD-L1 inhibition, and IL-15 stimulation [133]. Although srKD003 has limited application for tumors resistant to immune checkpoint inhibitors, its cognate, IL-15-based PD-L1-targeted immunocytokine LH01, can overcome primary resistance to PD-1/PD-L1 blockade by down-regulating TGF- levels within a tumor microenvironment, with no significant effect on PD-L1 expression levels. Additionally, it can induce local inflammation around malignancies, enhance CD8+ lymphocytes and NK-cell infiltration, and decrease the number of local T_reg_ populations [134].

At least one publication focuses on IL-17-derived immunocytokines. In this study, IL-17 was fused to the C-terminus of a scFv fragment of fibronectin EDA-specific F8 antibody to investigate the anti-tumor effects of the construct. The construct (F8-IL-7) was capable of binding both IL-17 receptors and fibronectin EDA in vitro. In vivo, F8-IL-17 stimulated angiogenesis and leukocyte infiltration in mice with subcutaneous F9 tumors, but no significant inhibition of tumor growth was observed [135]. Yet, more promising results were obtained with another proinflammatory cytokine, IL-21, a pleiotropic type I cytokine mainly produced by T and NK cells [136]. Using IL-21-based immunocytokines to improve anti-PD-1/PD-L1 therapy, as the lack of efficient T-cell activation may be responsible for low response rates to checkpoint blockaders in different cancer treatments, seems promising. Thus, an anti-PD-1 antibody may be used as a vehicle for delivering IL-21 to reactive T cells and, therefore, enhance their cytolytic properties. A fusion protein, PD-1Ab21, which comprises anti-PD-1 diabody and two IL-21 molecules, proved this assumption and showed potent antitumor effects in mice with established tumors. This effect was accompanied by an expansion of tumor-specific CD8+ T cells, and it increased the frequency of memory stem T cells superior to infusions of IL-21 combined with PD-1 blockader [137]. In humanized mice refractory to anti-PD-1, this approach provided significant protection [138].

## 9. Concluding Remarks

This article surveys the main immunocytokine formats, their modes of action, and properties, evaluating pre-clinical and clinical progress in this field. Due to their unique targeting properties, these molecules may aim at lesion sites, concentrate at the tumor microenvironment, and increase the therapeutic index of the cytokines. An improved serum half-life and tumor–to–blood ratio are other virtues that distinguish immunocytokines from the conventional cytokine therapeutics. Despite the promising results, only a few immunocytokines progressed to phase II, and even less moved to the subsequent steps. Being used as single agents in animal models, immunocytokines rarely show a complete tumor eradication; however, combinations of immunocytokines with other agents, including chemotherapy or monoclonal antibodies, more often result in a complete cure. Immunocytokines might deliver cytokines, which directly act on malignant cells, or locally activate immune system effector cells. For example, TNF-α-based therapeutics activate apoptosis, inhibit tumor cell proliferation, and disrupt tumor vasculature, whereas IL-2- or IL-12-based molecules activate and enhance NK and T cells. Interestingly, in some preclinical experiments, animals treated with immunocytokines demonstrated a stable vaccination effect and did not develop tumors after being challenged with the same cell line.

Irrespective of all the above-mentioned advantages of cytokines and their success, systemic toxicity intrinsic to cytokines is still a serious challenge for immunocytokine applications. Some immunocytokines (e.g., those containing IL-2) may be sequestered by high-affinity receptors or the excess of peripheral cells containing cytokine-specific receptors, thereby activating them and decreasing the target-mediated drug disposition [139]. This may result in a limited tolerability and implies restrictions on immunocytokine use. However, to overcome these distinct adverse effects, scientists have designed masked immunocytokines, activated by matrix metalloproteases or tumor-specific proteases [42,43,140]. Another strategy focuses on cytokine moiety engineering, with a purpose to decrease the affinity of the cytokines to high-affinity receptors expressed on normal cells [44,141]. A prolonged half-life of immunocytokines may be another reason for their limited achievable dosing compared to free cytokines. Moreover, if full-scale IgG is used to construct immunocytokines, normal cells may be affected and eliminated by CDC or ADCC; therefore, IgG4 or CDC and ADCC-attenuated IgG are required for the construction [109]. All the described drawbacks alone or in combination could make immunocytokine specificity irrelevant to biodistribution or efficacy [142].

Notwithstanding all the above-mentioned limitations of immunocytokine treatment and challenges in their development, these recombinant proteins are still a promising class of immunotherapeutics, with a growing body of evidence in favor of their advantages over free cytokines. New techniques appeared to reduce the systemic toxicity of immunocytokines, thus overcoming the limiting doses and preventing off-target cytokine–receptor interactions. Taking everything into account, we anticipate that immunocytokines will find an increasing use in clinical practice for combination therapy with other established therapeutics and methods.

## Figures and Tables

**Figure 1 pharmaceutics-15-00336-f001:**
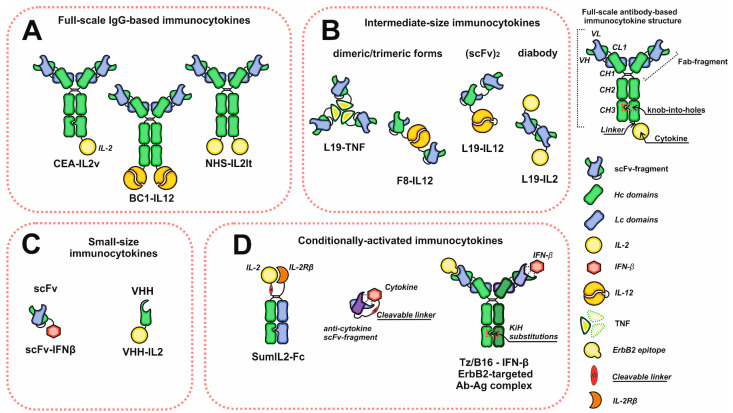
Schematic representation of the diversity of immunocytokine formats. (**A**) Immunocytokines based on full-scale IgG. (**B**) Intermediate-size immunocytokines. (**C**) Small-size immunocytokines based on antibody scFv or VHH fragments. (**D**) Conditionally activated immunocytokines. Abbreviations: VH and VL—heavy and light chain variable domains; CL—light chain constant domain; CH1, CH2, CH3—heavy chain constant domains; scFv—single chain variable fragment; VHH—single domain antibodies.

**Figure 2 pharmaceutics-15-00336-f002:**
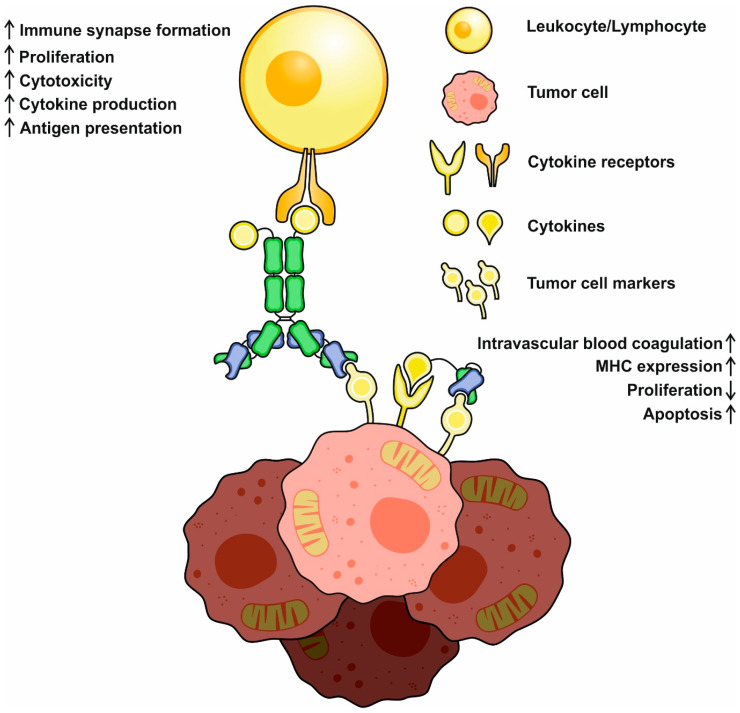
A schematic representation of basic mechanisms of immunocytokine action. Cytokines may activate and redirect immune system cells to tumor cells and initiate immune synapse formation. Activated immune cells increase (↑) in cytotoxicity, cytokine production, and proliferation. Direct action of cytokines decreases (↓) tumor cell propagation, increases MHC expression and antigen presentation, and might activate proapoptotic pathways.

**Table 1 pharmaceutics-15-00336-t001:** Immunocytokines for cancer treatment in clinical development. Fibronectin splice-isoforms and DNA/Histone complexes are the most frequent targets for the delivery of cytokines. IL-2-, IL-12-, and TNF-α-based immunocytokines are investigated in clinical trials.

Immunocytokine	Format	Target	Indications and Conditions for Treatment	Completed Clinical Trials	Clinical Trial Identifier, Ref.
CEA-IL-2v (cergetuzumab amunaleukine)	full-scale IgG	CEA	advanced and/or metastatic carcinoembryonic antigen (CEA)-positive solid tumors	Phase I	NCT02350673, [47]
L19-IL2 (Darleukin)	diabody	EDB	oligometastatic solid tumors	Phase I, comb. with radiotherapy	NCT02086721
			malignant melanoma (stages III and IV)	Phase II, comb with L19-TNFα	NCT02076633, [72]
			solid tumors and renal cell carcinoma	Phase I/II	NCT01058538, [73]
			malignant melanoma	Phase II	NCT01253096
F16-IL2 (Teleukin)	diabody	Tenascin–C	acute myeloid leukemia	Phase I, comb. with Anti-CD33 Ab	NCT03207191
			breast cancer, metastatic melanoma, non-small cell lung cancer	Phase Ib/II, comb. with paclitaxel	NCT01134250
NHS-IL-2Lt (Selectikine)	full-scale IgG	DNA/Histone complexes	lung cancer, non-small cell lung cancer	Phase I, comb. with radiotherapy	NCT00879866
			non-Hodgkin lymphoma	Phase I, alone and in comb. with cyclophosphamide	NCT01032681, [74]
hu14.18-IL2	full-scale IgG	GD-2	melanoma	Phase II	NCT00590824, [75]
			melanoma	Phase I/II, comb with radiotherapy, nivolumab or/and ipilimumab	NCT03958383
			skin melanoma, neuroblastoma, sarcoma	Phase I	NCT00003750
			recurrent or refractory neuroblastoma in children	Phase II	NCT00082758
huKS-IL2 (tucotuzumab celmoleukin)	full-scale IgG	EpCAM	prostate, colorectal and non-small cell lung cancers	Phase I	NCT00132522, [41]
			bladder cancer, kidney cancer, lung cancer	Phase I	NCT00016237
NHS-IL12	full-scale IgG	DNA/Histone complexes	epithelial neoplasms, malignantepithelial tumors, malignant mesenchymal tumor	Phase I	NCT01417546
BC1-IL12	full-scale IgG	EDB	metastatic renal cell carcinoma, metastatic malignant melanoma	Phase I	NCT00625768
L19-TNFα	scFv—trimeric form	EDB	solid tumors, colorectal cancer	Phase I/II	NCT01253837
			advanced solid tumors amenable to anthracycline, sarcoma, breast cancer, lung carcinomas	Phase I, combined with doxorubicin	NCT02076620
			melanoma (III and IV stages)	Phase I, combined with melphan (limb perfusion)	NCT01213732

## Data Availability

No new data were created or analyzed in this study. Data sharing is not applicable to this article.

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
