# Peer review of "Targeted Cytokine Delivery for Cancer Treatment: Engineering and Biological Effects"

_pharmaceutics, 2023, doi:10.3390/pharmaceutics15020336_

Round 1

Reviewer 1 Report

The authors provide an extensive overview of antibody-cytokine fusion proteins used in cancer therapy. The article is structured in a logical order. However, the language and style should be improved.

In the literature review, the authors neglect to cite a substantial and growing body of evidence that immunocytokines expose systemic target cells sufficiently to limit tolerability and achievable dosing (e.g., PMIDs 14528315, 25733854, 27611189, 29463551, 33986267, 34376502, 35286392, 35534574).

Reference 18 does not match the text.

Line 584: authors are describing an IL-17 fusion protein, however the reference 92 is on F8-IL7 and not F8-IL17.

Author Response

Dear reviewer, thank you for your deep and thoughtful analysis of the article and fair comments. Here I would like to respond systematically.

  • “Reference 18 does not match the text.”

 We corrected this reference.

  • “Line 584: authors are describing an IL-17 fusion protein; however, the reference 92 is on F8-IL7 and not F8-IL17”.

 Pasche N. has quite similar studies for F8-IL7 and F8-IL17 immunocytokines with a resembling design of experiments. In the text, we meant F8-IL17 and this reference:

 Pasche N., Frey K., Neri D. The targeted delivery of IL17 to the mouse tumor neo-vasculature enhances angiogenesis but does not reduce tumor growth rate. Angiogenesis 2012, 15, 165–169

We made following corrections both in the text and in the reference list.

  • In the literature review, the authors neglect to cite a substantial and growing body of evidence that immunocytokines expose systemic target cells sufficiently to limit tolerability and achievable dosing (e.g., PMIDs 14528315, 25733854, 27611189, 29463551, 33986267, 34376502, 35286392, 35534574)

Despite the growing body of evidence that immunocytokines expose off-target effects and systemic toxicities the number of articles, which affirm that the tolerability of immunocytokines and their advantages over cytokines alone, is higher. However, we acknowledge the importance of the mentioned problem and agree that it is not correct to completely omit this topic. So, after analysing given PMIDs we decided to add a new paragraph in the section 9.

  • We also made extensive grammar and language corrections through the whole text with a help of a professional specialist.

Best regards, respectfully,

Rybchenko Vladislav.

Reviewer 2 Report

The manuscript entitled "Targeted cytokine delivery for cancer treatment: engineering and biological effects" by Rybchenko et al. reviews the antibody-cytokine fusion protein formats, their targets, mechanisms of action, antitumor, and other biological effects. This is a well-written review manuscript appropriate for the Pharmaceutics journal. I have a couple of minor comments.

1.     Incorporating different cytokine delivery strategies, including nanocarrier-mediated cytokine delivery, would improve the overall quality of the review.

2.     It is highly recommended to add a table of various cytokine delivery systems under different phases of clinical trials.

Author Response

Dear reviewer, thank you for your deep and thoughtful analysis of the article, fair comments and evaluation of our manuscript. Here I would like to respond to your comments.

  • Incorporating different cytokine delivery strategies, including nanocarrier-mediated cytokine delivery, would improve the overall quality of the review.

We are informed about distinct strategies of targeted cytokine delivery, including nanocarrier-mediated ones. However, in the presented text we decided to focus on protein-based technologies, in general on antibody-cytokine fusion proteins or other muteins. The addition of other strategies will lead to significant text restructuring and change in the original concept of the article. That is why we prefer to avoid talking about that important delivery system and focus only on immunocytokines. However, we supplemented additional references and described conditionally-activated imunocytokine formats at the section 2.

  • It is highly recommended to add a table of various cytokine delivery systems under different phases of clinical trials.

The main focus of the review is the overview of the different structural formats of immunocytokines and the structure-function relations. A table of various cytokine delivery systems under different phases of clinical trials would be quite informative, although in our opinion it stays beyond of the scope of this review.

Following the reviewer’s recommendations we made extensive grammar and language corrections through the whole text with a help of a professional specialist.

Best regards, respectfully,

Rybchenko Vladislav.

Reviewer 3 Report

In this review article, the authors focused on actual antibody-cytokine fusion protein (immunocytokine} format, their targets, mechanism of action, anti-tumor, and other biological effects. They described related preclinical and clinical data and the efficacy of that antibody-mediated cytokine delivery processes either as a single therapy or in combination therapy. The review is based on their extensive survey and review of literature spanning many decades. Based on their review, they concluded that immunokines will continue to attract increasing attention regarding their clinical applications for immunotherapy either as single or combination therapy.

Overall, the article is nicely written and not difficult to follow. It has nice illustrations and models to support some relevant mechanisms. In addition, the article is relatively well-referenced. The content is valuable and represent extensive survey of literature in the immunotherapy field. The article is likely to attract readers and will add information to the field.

However, there are some concerns/weaknesses identified, which require the attention of the authors to address in order to further enhance the overall merit of the article.

Major concerns:

·         Even though the field on applications of specific cytokines including IL-2 in immunotherapy has been far advanced in United States through work supported by the National Cancer Institute (NCI), some of the key references were omitted from this review article. Also, some references for work done on immunotherapies in other parts of the world were not included. It could be that the authors have limited access to some of the vital publications from United Stated and other parts of the world. In any case, it is especially important that some of these groundbreaking work and advances be referenced in this article. Below are a few of such relevance references that should be cited in this article.

1.        S. A. Rosenberg (2014) IL-2 is the First Effective Immunotherapy for Human Cancer. J. Immunol 192 (12) 5451-5458.

Dr. Rosenberg is the pioneer on cancer immunotherapy using cytokines.

2.       John M. Wrangle et al. (2018). IL-2 and Beyond in Cancer Immunotherapy. J. Interferon & Cytokine Res. 38, # 2 https: doi.org/10.1089/jir.2017.0101

3.       Tao, J. et al. (2016). Role of IL-2 in Cancer Immunotherapy. Oncoimmunol. Doi, 10.1080/201 402x.2016.1163462

4.       Lorenzo Montana et al. (2018) Anticancer Therapy Employing IL-2 Cytokine Tumor Targeting Combination of Innate, Adaptive, and immunosuppressive cells in Anti-Tumor Efficacy. Front. Immunology

5.       Willem, W. Overwijk et al. (2021). Engineering IL-2 to Give New Life to T cell Immunotherapy. Ann Review Med. (72) .281-311

6.        Renren Yu. et al. (2022) Types Interferon-mediated tumor immunity and its role in immunotherapy. Cell. Mol. Life Sci.

7.       Ling Ni and Jian Lu (2018) Interferon-gamma in Cancer Therapy. Cancer Med. 7: 4509-4516

8.       Angela, M. Gocher et al (2022). IFN-gamma: teammate or opponent in the tumor microenvironment 22: 1-15

9.       Guo, J. et al. (2019) Engineering therapeutic antibodies with IFN-alpha for cancer immunotherapy. PLOS ONE. 14(8): e0219829  

Other noticeable concerns:
In the article, there are many stretches of sentences without references and make it difficult to determine whether such lengthy sentences without references are statement of the authors or they are referring to work published by other investigators.

For example.

Page 1, lines 29 to 31.: citation should be provided

 Page 1, lines 31 to 33: citation should be provided

Page 2, lines 41 to 42: additional citations are needed for the “Life-threatening side effects and toxicity or cytokine release syndrome”

Page 2, lines 54 to 58: citation should be provided

Page 3, lines 102 to 111: additional citations should be provided

Page 3, lines 113 to 115: citation is needed

Page 8, lines 192 to 209: need more citations on applications of IL-2 in cancer immunotherapy

Page 9, lines 234 to 241, more citations needed

Page 9, lines 242 to 265: more citations need

Page 9, lines 267 to 278: additional citations needed

Page 10, lines 300 to 308: additional citations needed

Page 11: please, make sure the” Pashe” citation is properly numbered, and that number matches the statement in the article.

 Minor type errors:

Page 3 line 102: change “have” to has

There are few more type errors/grammatical errors in the article that need to be corrected

Author Response

Dear reviewer, thank you for your deep and thoughtful analysis of the article, fair comments and evaluation of our manuscript. Here I would like to respond to your comments.

All the articles you suggested to cite were read, analyzed, and added to the reference list. Based on that new information we decided to expand our text with additional statements and phrases to the corresponding parts of the manuscript.

According to your recommendations we also added additional references to the corresponding parts to make easier to distinguish between our statements and the published data and direct citations.

Finally, we made extensive grammar and language corrections through the whole text with a help of a professional specialist.

Best regards, respectfully,

Rybchenko Vladislav.

Reviewer 4 Report

The authors review literature on approaches used for targeted delivery of cytokines for cancer treatment. Overall, the topic of the review is of reasonable interest. The writing needs improvement throughout the review.  I have the below comments for the authors.

1.     My major concern is that the authors need to indicate how this review is different from several other existing reviews on targeted cytokines. This should be stated explicitly in the introduction.

2.     Several statements in the review are unclear or poorly written, and need to be rephrased for clarity. Below stated are a few examples.

“After discovery of cytokines those discussions were taken to the next level.”

“This may lead to immunocytokine depletion because of its trapping by the excess 89 of accessible cytokine receptors and lack of necessary concentrations at the target site [14]. 90.”

“More 152 important role in antibody-cytokine fusions plays cytokine, as it is a delivered molecule.”

“In the light of the above blind 318 administration of IL-12 is not pursued nowadays neither in future, but….” 

“An interesting observation made by Pashe 323 and coauthors reveals that the format of immunocytokine is of a great impact on the 324 effectiveness of the fuse protein if the cytokine molecular weight is high or it consists of 325 two or more subunits, e.g. IL-12.” 

“From the one hand TNF-α 394 promotes inflammation and regulates cell survival, immunosuppression and can induce 395 invasion of neoplastic cells into surrounding tissues.”

3.     I have the following comments for the figures.

a.     Fig 1: Would be helpful if a key is included to describe the schematic i.e., what each shape/colored object means (as in Fig.2). Also, for “D) Bispecific 135 antibody – cytokine complex” - suggest including brief description of the format.

b.     Fig 2: Suggest including a brief description of the schematic in the figure legend in addition to the figure title.

4.     Suggest including a table(s) summarizing various cytokines, immunocytokine therapeutics, engineering strategies, key results, current clinical stage etc. 

5.     Also, the authors should think of possibly including any additional figures that may improve the review.

6. Authors should proofread to check for typos, grammar and language throughout the review.

Author Response

Dear reviewer, thank you for your deep and thoughtful analysis of the article, fair comments and evaluation of our manuscript. Here I would like to respond to your comments.

Opposite to other reviews in this field, in our review we also describe cytokine formats designed to prevent off-target interactions, thus diminishing potential side effects. Also we reviewed and cited the latest articles and sources of information.

Following to your recommendation, we added a key in the figure 1 similar to figure 2. We also extended figure 1 by additional immunocytokine formats.

The description of the figure 2 was expanded by additional explanations of this figure.

The main focus of the review is the overview of the different structural formats of immunocytokines and the structure-function relations. A table of various cytokine delivery systems under different phases of clinical trials would be quite informative although in our opinion it stays beyond of the scope of this review.

Finally, we made extensive grammar and language corrections through the whole text with a help of a professional specialist.

Best regards, respectfully,

Rybchenko Vladislav.

Round 2

Reviewer 1 Report

The authors addressed all issues in the manuscript.

Author Response

Dear reviewer, we are thankful for your remarks and comments.

Best regards,

Rybchenko Vladislav.

Reviewer 3 Report

Dear Authors:

I am happy that you have addressed the identified concerns raised by the reviewers of your original submission. Notably, you have included the suggested additional citations, in both the body of the article and in the references' section. These, these actions have helped in improving the quality of your proposal. 

Author Response

Dear reviewer, we appreciate your remarks and help.

Sincerely yours,

Rybchenko Vladislav.

Reviewer 4 Report

I have the follwing comments for the authors.

1.     Fig 1 and 2 – The revisions address the previous comments. I appreciate the authors efforts to improve the figures.

a.     Revised Fig 1 looks much improved. However, suggest swapping C with D i.e, making “conditionally-activated” as sub-figure D in Fig 1.

2.     As stated earlier, there are several other existing reviews on the topic ‘targeted cytokines’. I am not convinced with the authors response (stated below) to earlier comment in this regard.

“Opposite to other reviews in this field, in our review we also describe cytokine formats designed to prevent offtarget interactions, thus diminishing potential side effects.”

3.     Several statements in the review are unclear or poorly written, and need to be rephrased for clarity. While the authors seem to suggest that they have rendered the services of a professional specialist, there are still several statements needing attention. Below stated are a few examples.

“The discovery of cytokines took these discussions to the next level [1 - 5] and gave the posibility of cytokines use in clinical practice.

“demonstrated promising antitumor activity with significant percentage of patient 242 response to therapy”

“A few other lines of 245 evidence make the clinical application of this cytokine controversial.”

“In several experiments 546 involving anti-PD-L1 resistant singeneic mouse tumor models IFN-α-antibody fusions 547 were tollerated well and demonstrated anti-tumor efficacy [112], which holds a promise 548 for therapy in case of anti-PL-L1 treatment failure.”

“One of them implies using a neutralizing bispecific 591 antibody with two specificities, the first one to the tumor cell marker and the second one 592 to interferon.”

4.     As pointed out previously, the authors should carefully proofread to check for typos, grammar and language throughout the review. Below are only a few examples of misspelt words and by no means is an exhaustive list.

a.     Line 35 – posibility

b.     Fig 2 legend – espression

c.     Line 380 – fuse protein

d.     Line 547 – singeneic

e.     Line 548 – tollerated well

f.      Line 549 – anti-PL-L1

g.     Line 597 -  metalloproteinase

5.     Suggest defining abbreviations on first mention e.g., Treg.

6.     The following comment was made previously: “Suggest including a table(s) summarizing various cytokines, immunocytokine therapeutics, engineering strategies, key results, current clinical stage etc.” 

a.     The author’s response seems to be vague and does not address the comment appropriately. A table summarizing different structural formats of immunocytokines and the structure-function relationship that has been discussed in the text of this review, along with the current clinical phase of testing would certainly improve the quality of the manuscript.

Author Response

Dear reviewer, we are thankful for your remarks and comments. Here we list following corrections made by our team.

  1. Your suggestions concerning figure 1 have been fulfilled.
  2. In our opinion, this review might be considered as an update of the scientific information on immunocytokines with the special attention to the cytokine formats designed to prevent off-target interactions, thus diminishing potential side effects.
  3. All the phrases mentioned by you were rewritten and checked by professional translators and correctors.
  4. Grammar and language were checked and corrected by professional translators and correctors Gavrilova and A. Sharapkova (Rosetta Stone MSU).
  5. We added abbreviation definitions at first mention for most of non-common terms (e.g. Tregs, CDC, ADCC)
  6. We also added a table summarizing anti-cancer immunocytokines which completed distinct phases of clinical trials.

Best regards,

Rybchenko Vladislav.

Round 3

Reviewer 4 Report

I have no further comments for the authors.